# Improved Medication Adherence of an Elderly Diabetic Patient at a Dwelling Home Using a Pill Dispenser and Personal Health Records

**DOI:** 10.3390/healthcare12040499

**Published:** 2024-02-19

**Authors:** Ryoji Suzuki, Emiri Takahashi, Ikuo Tofukuji

**Affiliations:** Department of Healthcaere Informatics, Faculty of Health and Welfare, Takasaki University of Health and Welfare, Takasaki City 3700033, Japan

**Keywords:** pill dispenser, personal health record, medication adherence, elderly, dwelling home

## Abstract

Fookkun^®^ is a pill dispenser in which single doses of several medications intended to be taken simultaneously are sealed in single film bags rolled onto a medication rotating drum. The system makes musical alert sounds when it is time for the patient to take the medications. If the patient misses a dose, a designated contact, such as the patient’s child, is alerted. We conducted an experiment monitoring the use of a pill dispenser (Fookkun^®^) by an older patient. The participant was a 71-year-old woman with diabetes living in a dwelling home. The experiment lasted approximately 6 months. Fookkun^®^ and the prototype data transmitter were installed at the patient’s home. Fookkun^®^’s medication history data are displayed on the electronic medication record book (E-MRB) and the patient’s pharmacist checks the patient’s medication history on the E-MRB. The Fookkun^®^ was effective in facilitating medication adherence. The pharmacist and the patient’s daughter did not need to check the E-MRB because Fookkun^®^ alerted them when the patient missed her medication. We believe that if the medication history data linked between a pill dispenser and an E-MRB can be shared among medical staff, this will contribute to a medical digital transformation in Japan in the future.

## 1. Introduction

In Japan, 36.24 million people are aged 65 and over, representing an ageing rate of 29.0%. Among these, 7,427,000 (28.8%) live alone [1]. Solitary deaths have become a social problem, underscoring the need for information-and-communication-technology-based support to monitor people.

Elderly patients residing at home often forget to take their medications. The value of unused medications in Japan is EUR 320 million [2]. Terauchi et al. reported that approximately 33% of type-2 diabetic patients who completed their questionnaire had unused medications and higher HbA1c levels. Therefore, a one-dose package (ODP) is recommended and managed by families and medical and care staff [3]. Onda et al. reported that when pharmacists visited patients’ homes twice per month, 29.8% of the patients showed improved adherence compared to that before the onset of home visits [4]. Physicians and pharmacists recommend using ODPs for patients who experience difficulty remembering to take medications, and community pharmacists also recommend using a weekly medication calendar. 

In a previous study, we developed and evaluated an ODP medication support system (ODP-MSS). ODP-MSS is a pill dispenser in which single doses of several medications intended to be taken at the same time are sealed in single film bags that are rolled onto a rotating drum. With this system, a musical alert sounds when it is time for the patient to take their medication. If the patient misses a dose during the set period due to forgetfulness, a voice message stating that the patient did not take their medication is sent to their medication supporters, such as the patient’s child, via telephone. ODP-MSS units were provided to 10 elderly patients living at home, with adherence assistance provided by family members or other medication supporters in response to telephone alerts. The findings of this study confirmed the usefulness of the ODP-MSS and ODP-MSS-directed follow-up calls from a medication supporter (as a system of telecare home monitoring) in reducing missed doses owing to forgetfulness [5]. The ODP-MSS has since been improved. In 2015, the medication administration support device with a monitoring function, ‘Fookkun^®^’ (Ishigami Manufacturing Co., Ltd., Iwate, Japan), was launched. 

Previdoli et al. demonstrated that, in most cases, the aim is to improve older people’s adherence and increase their knowledge of the prescribed medication [6]. However, most studies involve one-to-one educational or coaching attempts. Comparatively, the approach in our study is distinct as it assesses and increases patients’ adherence to prescribed medication using ODP-MSS.

Three similar ODP medication support devices exist. Specifically, Rantanen et al. examined the safety profile and use of an integrated advanced robotic device (Evondos E300, Evondos Telecare System, Salo, Finland). Twenty-seven home-dwelling patients retrieved their medicine sachets for 99% of the alerts. All patients and 96% of the nurses reported that the device was easy to use [7]. Patel et al. reported that 58 participants (mean age of 66.36 years) and 11 caregiver participants used ‘Hello I’m Spencer’ (Catalyst Healthcare Ltd., West Hollywood City, CA, USA). The caregiver burden before and after using the smart medication dispenser for 6 months showed a statistically significant difference (*p* < 0.001) [8]. Hannink et al. reported that 36 patients (mean age = 69 years) with Parkinson’s disease used the Medido Connected (Innospense BV^®^, The Hague, The Netherlands). The use of the Medido was thought to potentially result in a clinical improvement in physical disability and seemed particularly appropriate for patients in more severe states [9].

Devices other than ODP medication support devices include automatic pill dispensers (with 28 exceedingly small compartments), which have been found to improve medication adherence in elderly patients with mild cognitive impairments and in patients with chronic heart failure [10,11]. The MD.2 medication dispenser (e-pill Medication Reminders, Wellesley, MA, USA) contains 42 cups for various dosage times and requires refilling approximately every 2 weeks [12,13,14]. Some of the desirable features that pharmacists and caregivers value include product simplicity, portability, options to lock the product, and the ability to assist with drug inventory management. These products might allow patients to independently manage their medications and could benefit highly motivated patients interested in taking control of their health and younger older adults who are more familiar with the technology [15]. However, highly motivated patients and younger older adults may not want to use such products as they have the potential to harm their dignity because these devices are locked. With the exception of Fookkun^®^, all medication support devices have a lock. Additionally, older people consider these devices to be difficult to refill, potentially making them cumbersome to use.

In Japan, the digital transformation (DX) of medical information is challenging, and the use of personal health records (PHRs) is not widespread. Paper medication record books (P-MRB) record prescription information; they became popular after the Great East Japan Earthquake in 2011. Patients give their P-MRBs to the physician when they visit the hospital, and the physician checks the prescription history data to ensure that there are no duplicate prescriptions. These data are also given to the pharmacist when the patient visits the pharmacy, and the pharmacist checks for errors in the prescribed medications. Over 40 types of electronic medication record books (E-MRBs) exist. Most E-MRBs have an alarm function for the medication administration time and the ability to manually input the medication taken (or not taken). However, links between the data reported by medication support devices and the medication history contained in E-MRBs do not exist. Additionally, the Ministry of Health Labour and Welfare (MHLW) promotes the use of E-MRBs (one type of PHR in Japan) [16]. Nevertheless, their use is only partially widespread. 

Andrikopoulou et al. reported that the National Health Service policy assumes that increasing the usage of PHRs by citizens will reduce the demand for healthcare services. Limited evidence exists indicating that health apps can improve patient outcomes [17]. Furthermore, 13 studies reported that using a PHR increased medication adherence [18]. The available methods for measuring adherence can be categorized as direct (e.g., directly observed therapy) or indirect (e.g., electronic medication monitors) [19]. Dasgupta et al. showed that the complex medical regimen maintained by many seniors frequently makes direct medication entry via an application difficult [20]. However, medication adherence and PHRs face the following challenges: (1) medication support devices (pill dispensers) can improve adherence, but their data are not linked to the PHR, (2) physicians or pharmacists need to directly inquire if patients have taken their medication, and (3) patients need to input their medication intake into the PHR; (4) therefore, the PHR lacks accurate data on medication adherence.

To address these issues, this study aimed to monitor an elderly patient using a pill dispenser (Fookkun^®^) and a PHR (prototype E-MRB) to verify the patient’s medication history with their pharmacist. We also hypothesized that improved medication adherence would result in lower HBA1c values.

## 2. Materials and Methods

### 2.1. Study Design

Fookkun^®^ is a pill dispenser in which single doses of several medications intended to be taken simultaneously are sealed in single film bags rolled onto a medication rotating drum. The system can dispense a maximum of 3 ODP doses daily for 60 days (Figure 1). To dispense more than three doses a day, an additional unit allows the device to dispense up to six doses a day. The system makes musical alert sounds when it is time for the patient to take the medications. If the patient misses a dose owing to forgetfulness during the set period (30 min), a voice message stating that ‘The medicine has not been taken’ is sent via telephone to one or more of the patient’s medication supporters, such as their son or daughter or pharmacist. The system also allows medication supporters to assess the patient’s health status over the telephone. The patient’s medication adherence can be determined by examining the internal memory [1] (Figure 2a).

Prototype data transmitter: A prototype data transmitter was developed to retrieve Fookkun^®^’s medication history data. The prototype data transmitter performed the following functions: (1) The universal serial bus cable was connected to the Fookkun^®^ to retrieve data on the identity, date, and time the medication was taken, and whether or not the medication was taken from the internal memory. (2) The data were then sent to the cloud server of the E-MRB ‘Hoppe^®^’ (Hoppe Co., Ltd., Tokyo, Japan) (once per hour) via a Long-Term Evolution (LTE) communication line (Figure 2b).

Prototype E-MRB: The prototype function in Hoppe^®^ had the following functions: (1) data from a data transmitter were received using a fixed IP address once per hour, and (2) features to evaluate the medication history were available to the pharmacists (‘pharmacy viewing system screen’) and patients/their families (‘viewing system screen in the smartphone’.

Data on medication (taken and not taken) were retrieved from Fookkun^®^, and the reception of data from the prototype data transmitter was confirmed by Hoppe^®^. Figure 3 shows a snapshot of the pharmacy viewing system screen for pharmacists. The vertical axis represents the date, while the horizontal axis represents the medication time (morning, noon, and evening). When a patient takes a medication, the time and symbol ‘o’ are displayed; if the patient forgets to take the medication, the time and the symbol ‘x’ are displayed. Also, if the patient is not prescribed a medication to be taken at noon, the symbol ‘x’ is displayed. The pharmacist can check the patient’s medication adherence by reviewing the pharmacy viewing system screen.

### 2.2. Study Participant

The participant was an elderly patient with diabetes residing in a dwelling home. The inclusion criteria were as follows: patients who frequently forget to take their medications and cannot adequately manage their medications by themselves, patients who cannot control their HbA1c levels, patients who take ODP medications, and patients who can take their medications independently or with minor assistance. The exclusion criteria were as follows: patients who are not taking ODP medications and have dementia.

### 2.3. Evaluations

The monitoring experiment was conducted from August 2022 to September 2023. We installed the Fookkun^®^ and the data transmitter in the patient’s home. At the beginning of the experiment, the patient was asked to fill out a questionnaire with their medication information. At the end of the experiment, the patient, their medication supporters, and pharmacists were asked to fill out a questionnaire about instances of missed doses owing to forgetfulness and the usefulness of the Fookkun^®^ and the prototype E-MRB. Further, the pharmacists were asked to provide the patient’s HbA1c routine test results. 

This study was conducted according to the Declaration of Helsinki guidelines and was approved by the medical ethics committee of Takasaki University of Health and Welfare (No. 2164, 11 March 2022). The consent forms for the study were received from the patient, their medication supporters, and pharmacists.

## 3. Results

### 3.1. Monitoring Experiment

The participant was a 71-year-old woman with diabetes in Takasaki City, Gunma Prefecture, selected by the pharmacist at the community pharmacy. The pharmacist at the community pharmacy was introduced to the board members by the Gunma Pharmaceutical Association. Table 1 lists the demographic and baseline medication data. Before the experiment, a pharmacist visited the patient once a week to set up a weekly supply of medications on a medication calendar. However, she forgot to take her medications once or twice after dinner on weekends. The patient’s two daughters and a pharmacist were chosen as medication supporters. The Fookkun^®^ device calls the patient’s medication supporters in turn when the patient forgets to take her medications.

From the questionnaire results provided by the pharmacist, patient, and patient’s daughters, it is inferred that Fookkun^®^ is effective in motivating adherence to medication and eliminates instances in which the patient forgets to take medicine. However, the pharmacist said that on three occasions, the patient had opened the door of the Fookkun^®^ to retrieve her medications, and that the medications had fallen into the device. Therefore, medication adherence improved when the Fookkun^®^ was used. Additionally, the pharmacist indicated that there was no need to check for remaining medications. The pharmacist and the patient’s daughter did not need to check the E-MRB because Fookkun^®^ called them when the patient forgot to take her medications. The patient and her daughter indicated that they liked the use of Fookkun^®^ because it became less likely that the patient was going to forget to take her medications (Table 2 and Table 3). The HbA1c values in the studied patient remained unchanged before and after the experiment.

### 3.2. Patient’s Life-Threatening Episodes

On 14 August at 18:30, the patient’s two daughters and a pharmacist were called in turn by the Fookkun^®^ as the patient did not take her medications, but none of them were able to answer immediately. The eldest daughter called the patient’s home, but the patient did not respond. A home-care nurse visited the patient’s home and entered the room with a duplicate key, but the patient was not present. After receiving a report from the home-care nurse, the daughter called the neighbors and found out that the patient had fallen ill in the supermarket and was admitted to the hospital by ambulance. The eldest daughter reported that she was grateful Fookkun^®^ allowed her to be notified of the patient’s safety from afar and avoid worse outcomes.

Figure 4 shows the E-MRB data from when the patient fell ill. We subsequently checked the E-MRB by smartphone, which displayed the symbol ‘o’ in the morning of 14th August, the symbol ‘x’ from the evening of the 14th until the morning of the 16th, and the symbol ‘o’ in the evening of the 16th. No medications were taken at noon; therefore, the symbol ‘x’ is shown. We found out that the patient had left her home in the morning of the 14th and felt unwell. Additionally, we discovered that the patient was discharged and returned home during the day on 16th August.

## 4. Discussion

### 4.1. Monitoring Experiment of a Diabetic Patient

In this study, we found that (1) the follow-up calls from Fookkun^®^ plus to a medication supporter helped prevent missed doses, improving the patient’s medication adherence, (2) the home pharmacist did not need to check for remaining medications, (3) the pharmacist and the daughter did not need to check the E-MRB, (4) and the HbA1c values remained unchanged.

Our previous study reported that using the ODP-MSS for 3 months in four hypertensive patients resulted in 100% medication adherence during the experimental period but yielded no significant differences in blood pressure before or after the experiment [21]. In this study, the target population was changed to a diabetic patient. However, the diabetic patient’s HbA1c values remained unchanged before and after the experiment over the 6-month study period. It was thought that the long-term use of Fookkun^®^ may be necessary to confirm changes in vital signs (blood pressure, HbA1c, etc.) owing to improved medication adherence. 

A pharmacist visited the patient’s home once a week before the experiment to set the medication on the medication calendar and check for remaining medications. This study reveals that the pharmacist no longer needs to check for remaining medications in the patient’s home when Fookkun^®^ is used. Therefore, the use of Fookkun^®^ for patients in dwelling homes is expected to reduce the time that the pharmacist’s spends setting up medications and checking for remaining medications.

In another study using similar ODP medication support devices, the Evondos E300 was used for 727 days by 27 patients (mean: 26.9 d per patient). Notably, 98.7% of the alerts resulted in patients retrieving their medicine sachets on time [7]. In another observational study, the monthly adherence rate remained >95% over the course of 6 months among the 46 participants using ‘Hello I am Spencer’ [8]. As in our study, these devices were effective in improving medication adherence.

However, similar devices have locks on them, and all medication administration must be performed by medical staff. Turjamaa et al. used multiple reminders and reported that, despite these reminders, the medication was not taken, the dose remained locked inside the device, and home care professionals were notified [22]. The reason that Fookkun^®^ is not locked is attributed to the experience of the Great East Japan Earthquake of 2011. Residents did not evacuate their homes with their medications or P-MRB at the time of the disaster, and some patients became ill in subsequent days as they were unable to take their medications at the shelter. Patients using Fookkun^®^ can remove the medication rotating drums and flee their homes in the event of an earthquake or power outage and can thus continue to take their medications. 

In our previous study, one patient shifted from family- to self-managed medication administration after using the ODP-MSS [4]. However, in this study, once the patient became accustomed to Fookkun^®^, they sometimes opened the lid to remove the medication. Therefore, Fookkun^®^ is not considered for use by patients with advanced dementia.

In this study, the pharmacist did not use the E-MRB medication history data very often. A pharmacist was one of the medication supporters in this study, and Fookkun^®^ would call the pharmacist when the patient did not take their medications. In addition, the pharmacist frequently exchanged the patient’s information with her family members and visiting nurses via medical SNS (MedicalCare STATION, Embrace Co., Ltd., Tokyo, Japan). If pharmacists do not become medication supporters and do not use medical SNS, the only way to track medication history is through E-MRBs. Accordingly, the use of E-MRBs is expected to increase even more. Takamatsu et al. reported that 547 pharmacies used a P-MRB, but only 15.9% of them used an E-MRB with a P-MRB [23]. Therefore, the widespread use of E-MRBs is a challenge. It is thought that pharmacists could improve medication adherence by checking E-MRBs more frequently in patients with low adherence.

### 4.2. Data Link of Pill Dispenser and PHR

One challenge with medication information is that there are no historical data on patients’ medication history (whether they took or did not take their medications). No published studies have linked the medication support device, pill dispenser (Fookkun^®^), and PHR (Hoppe^®^) data. To our knowledge, this is the first study to address this issue. 

Boland et al. reported the use of PHRs to store a list of patient once-daily glaucoma medications and reminder preferences. The adherence rate in the 20 participants in the intervention group increased from 54% to 73% (*p* < 0.05), but no statistical change was observed in the 19 participants in the control group [24]. Chen et al. reported that the 46 atrial fibrillation patients who used the PHR became more knowledgeable about their medications, resulting in improved medication adherence [25]. 

Furthermore, Chrischilles et al. reported the results of giving eligible computer users aged 65 and access to a PHR (*n* = 802). A majority (55.2%) logged into the PHR and used it, but only 16.1% used it frequently. No difference was observed in the use of inappropriate medications or adherence measures [26]. Andrikopoulou et al. found recurrent evidence that PHRs can improve medication adherence, but little evidence indicating which design features facilitate this process is available to date [17].

In Japan, the MHLW has issued E-MRB guidelines [16]. E-MRBs are required to be able to check the patient’s medication schedule, notify the user when medication should be taken, and record the time at which the medication has been taken. In the future, it is envisioned that a function will be implemented to automatically calculate and display the number of remaining medications based on the medication schedules and dose records. Therefore, we believe that the PHR could be used by elderly patients effectively if it has a medication time alarm function and can automatically input medication history data from the pill dispenser.

In Finland, home care has been encouraged to implement robotic medication management, as the use of robots for medication management allows the implementation of older people’s independent medication management [27]. Turjamaa et al. reported the use and competence needs of a robot for medication management in older people’s home care. The successful implementation and use of the robot for medication management required a timely and adequate introduction before the implementation of the robot, the easy and practical use of the robot in daily work, and confidence in work competence. Home-care professionals need to re-organize the use of digital solutions to make their workflow smooth and easy, and prevent burnout and turnover among home-care professionals [22]. 

Conversely, in Japan, medical DX is defined as the standardization of the data generated in health, medical, and long-term care to promote disease prevention and to change the shape of society and life so that people can receive better-quality medical care [28]. This is the first attempt in Japan to link medication history data between a pill dispenser and a PHR. We believe that if the medication data of the patient can be shared among medical staff, this would contribute to medical DX in Japan in the future.

A dearth of research exists on how robots might change geriatric care and the work environment within assisted living facilities. Trainum et al. reported that care robots constitute an intervention that has the potential to improve the care of older adults and the work life of their professional caregivers [29]. In our previous study, we developed a drug distribution support device for caregivers, which was subsequently installed in a group home, and a 3-month monitoring experiment was conducted. The drug distribution support device reduced the daily dispensing duration by an average of 3.5 min [30]. Therefore, the introduction of pill dispensers at the facility is expected to reduce the labor of the caregivers.

### 4.3. Study Limitations

There was only one participant in this study. The reasons for this are as follows: (1) there was a period when patients could not be selected because they could not be visited at home owing to the coronavirus 2019 pandemic, (2) the study was limited to diabetic patients at home, patients who frequently forgot to take their medications, and could not adequately manage their medications by themselves.

In the future, we intend to study the medical evidence of pill dispenser. Therefore, with the cooperation of the university hospital, we intend to conduct experiments on patients with lifestyle-related diseases (hypertension, diabetes, and lipid abnormalities), categorising them into two groups: the Fookkun^®^ use group and the control group. We also aim to contribute to Japan’s ageing society by linking pill dispenser and PHR medication history data through industry-academia collaboration.

## 5. Conclusions

To our knowledge, this is the first attempt in Japan to link medication history data between a pill dispenser (Fookkun^®^) and an E-MRB (Hoppe^®^). The results of the monitoring experiment (for approximately 6 months) in a diabetic 71-year-old woman were reported herein. The findings revealed that Fookkun^®^ is effective in motivating adherence to medication. In this study, the patient’s pharmacist did not need to check the E-MRB because Fookkun^®^ called when the patient forgot to take her medications. We believe that if the medication history data linked between a pill dispenser and an E-MRB can be shared among medical staff, this will contribute to medical DX in Japan in the future.

## Figures and Tables

**Figure 1 healthcare-12-00499-f001:**
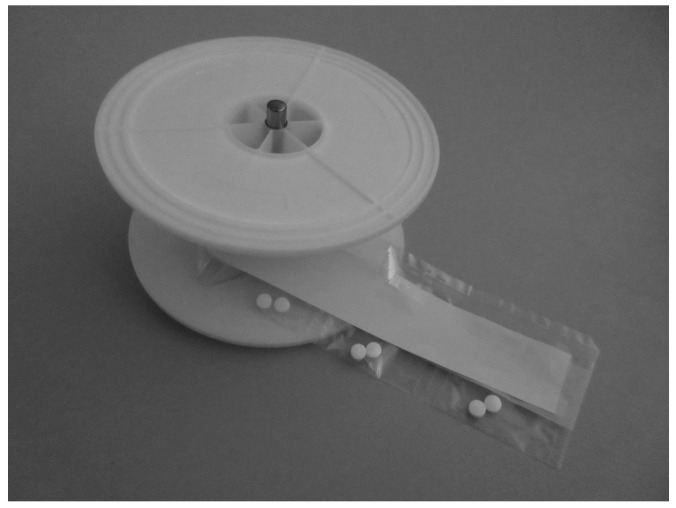
Fookkun^®^’s medication rotating drum. (1000 × 750 pixels).

**Figure 2 healthcare-12-00499-f002:**
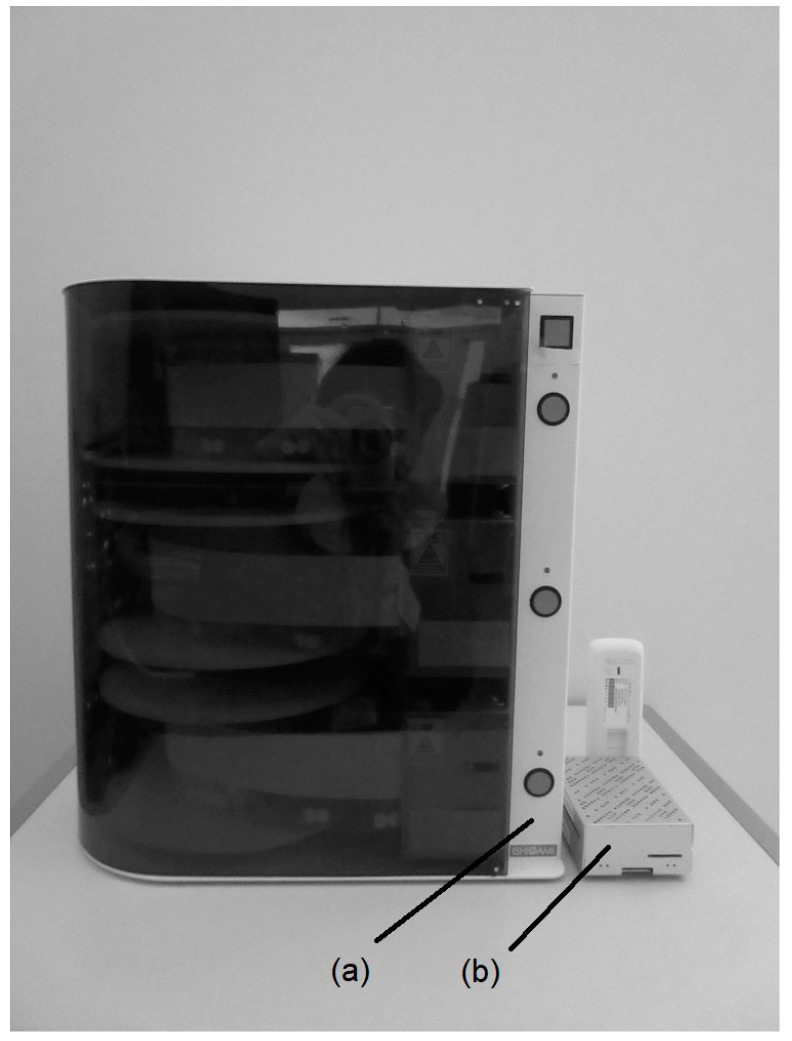
(**a**) Medication administration support device with the monitoring function ‘Fookkun^®^’. (**b**) Prototype data transmitter. (750 × 1000 pixels).

**Figure 3 healthcare-12-00499-f003:**
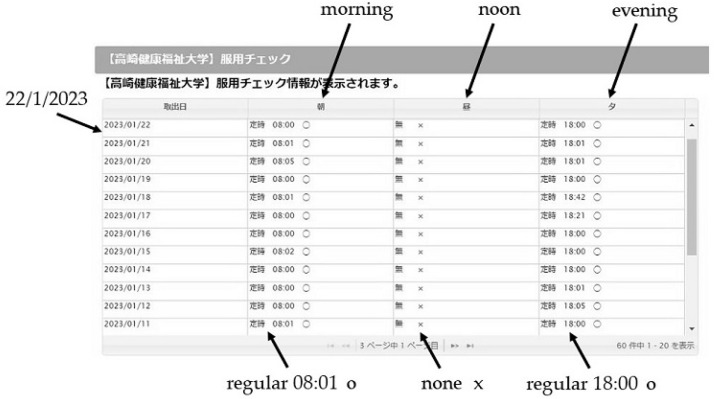
Pharmacy viewing system screen (850 × 450 pixels).

**Figure 4 healthcare-12-00499-f004:**
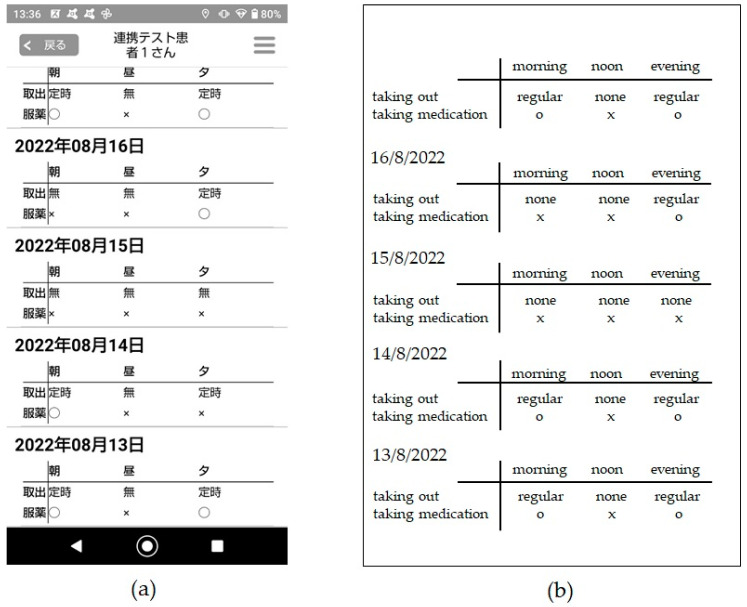
Data of E-MRB when the patient had fallen ill: (**a**) Smartphone screen (in Japanese); (**b**) Screen image modified in English (800 × 660 pixels).

**Table 1 healthcare-12-00499-t001:** Demographic and baseline medication data.

Item	Data
Sex	Female
Age	71
Living arrangement	Alone
Home care services	care worker, home nurse, home pharmacist
Disease	diabetes, heart disease, bipolar disorder
Medication frequency	after breakfast, after dinner
Medication management	home pharmacist/week
Medication tool	medication calendar
Forgot to take the medicine	1–2 times/after dinner on weekends
Medication supporters receive phone calls from Fookkun^®^	first: eldest daughter second: second daughter third: home pharmacist
Value of HbA1c	7.2

**Table 2 healthcare-12-00499-t002:** Pharmacist questionnaire results.

Questionnaire	Answer
1. Did the Fookkun^®^ help the patient remember to take it?	Useful
2. What has changed compared with your previous medication guidance before and after using Fookkun^®^ and E-MRB?	No need to check for the remaining medication
3. Was the medication history data from the E-MRB useful for medication guidance?	Not used
4. Do you think checking medication history is effective in preventing infection?	Slightly more effective
5. Would you recommend the Fookkun^®^ to other patients?	Yes
6. Would you recommend other pharmacists use the medication history data from the E-MRB?	I cannot answer that question because I do not use it.
7. Others	(1) Family members and medical personnel could share information about urgent changes in patients. I think this is effective for patients who live alone (2) Once patients became accustomed to the Fookkun^®^, they sometimes opened the lid to remove the medication (3) Fookkun^®^ is effective in facilitating medication adherence

**Table 3 healthcare-12-00499-t003:** Questionnaire results obtained from the patient and her daughter.

Questionnaire	Patient’s Answer	Daughter’s Answer
1. Did you forget to take medication during the experiment?	On some occasions	My mother got into the habit of taking her medications; thus, forgetting to take them has been prevented
2. If you forgot to take your medication after receiving a phone call from a supporter, did you take your medication or not?	After receiving a phone call from a daughter, I took the pills	Calling my mother when she forgot to take her medication was not burdensome
3. Do you want to use Fookkun^®^ in the future?	I would like to use Fookkun^®^ because it would means that I am less likely to forget to take medicine	My mother used to forget to take her medication, but Fookkun^®^ was useful because she remembered to take it more often
4. Did you check the E-MRB?	I am not using a smartphone	I did not check the E-MRB because Fookkun^®^ called me when my mother forgot to take her medication

## Data Availability

The author confirms that the data supporting the findings of this study are available within the article.

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
