# Peer review of "Improved Medication Adherence of an Elderly Diabetic Patient at a Dwelling Home Using a Pill Dispenser and Personal Health Records"

_healthcare, 2024, doi:10.3390/healthcare12040499_

Round 1
Reviewer 1 Report
Comments and Suggestions for Authors
Thank you for the opportunity to review this manuscript. The authors have, without a doubt, a large amount of knowledge within the field investigated, but at times, the knowledge is presented in a way that readers without specific knowledge within the field can have difficulties understanding the text.
There are many abbreviations in the text; either should this be reduced or a list of the abbreviations should be added.
Abstract
Line 9 - 11. I don’t understand the connection between the systems that are used.
I don’t think this sentence is needed: To our 14 knowledge, this is the first attempt in Japan to link medication history data generated by a pill dis-15 penser and an E-MRB.
Introduction
Line 37: Please, explain shortly what the ODP-MSS consists of.
Line 48: “the medical support system” this part could refer clearer to the aim of increasing adherence to prescribed medication – this is what you mean?
Line 50 – 59: Please be clearer on the systems you present and the results from studies on them. Do you present 2 or 3 supporting systems?
Line 66: rephrase the part “of they feel”
Line 72: “P-MRBs are to the physician by the patients…” A word must be missing here.
Line 80: Delete the abbreviation NHS. The are many abbreviations in the text and this one is not needed.
Line 97: Does this mean that patients who need medication 4 times daily cannot use the system?
Line 106: What is a data transmitter?
Figure 3: Does the “x” in the middle of the figure mean that the patient hasn’t taken the medication in the middle of the day at any day?
Line 133: Delete “do not have dementia” This is repetition from exclusion criteria.
Line 134: Delete “patients who are not taking ODP medications, do not forget to take their medications, have full medication assistance…” This is repetition from inclusion criteria.
Line 153: On line 136 you write that the study was conducted from August 2022 to September 2023, here your write from August 8, 2022, to February 13, 2023. Please, describe one period at one place only.
Line 175 Table 4. I don’t think the table is needed.
Line 179 – 197: These sections are too detailed and should be shortened.
Line 200: It would be beneficial if the discussion started with a short resume of the findings of this study.
Line 215 – 225: Please, relates these results to your study.
Lie 245: Do you need to describe this that detailed?
Line 298: Do not introduce a new abbreviations (DDSD) for something you only use twice.
Author Response
Response to Reviewer 1
Thank you for the favorable peer review. With regard to your comments, we would like to respond as follows.
Abstract
Line 9 - 11. I don’t understand the connection between the systems that are used.
In response to your suggestion, we have made the following added as follows: Fookkun® is a pill dispenser in which single doses of several medications intended to be taken simultaneously are sealed in single film bags rolled onto a medication rotating drum. The system makes musical alert sounds when it is time for the patient to take the medications. If the patient misses a dose, a designated contact, such as the patient’s child, is alerted. We conducted an experiment monitoring the use of a pill dispenser (Fookkun®) by an older patient. Fookkun®’s medication history data is displayed on the electronic medication record book (E-MRB) and the patient’s pharmacist checks the patient's medication history on the E-MRB.
I don’t think this sentence is needed: To our 14 knowledge, this is the first attempt in Japan to link medication history data generated by a pill dis-15 penser and an E-MRB.
We deleted the sentence as you pointed out.
Introduction
Line 37: Please, explain shortly what the ODP-MSS consists of.
In response to your suggestion, we have made the following added as follows: ODP-MSS is a pill dispenser in which single doses of several medications intended to be taken at the same time are sealed in single film bags that are rolled onto a rotating drum. With this system, a musical alert sounds when it is time for the patient to take their medication. If the patient misses a dose during the set period due to forgetfulness, a voice message stating that the patient did not take their medication is sent to medication supporters such as the patient’s child via telephone.
Line 48: “the medical support system” this part could refer clearer to the aim of increasing adherence to prescribed medication – this is what you mean?
In response to your suggestion, we have made the following added as follows: Comparatively, the approach in our study is distinct as it assesses and increases adherence to prescribed medication using ODP-MSS.
Line 50 – 59: Please be clearer on the systems you present and the results from studies on them. Do you present 2 or 3 supporting systems?
In response to your suggestion, we have made the following corrections: Three similar ODP medication support devices exist. Specifically, Rantanen et al. examined the safety profile and use of an integrated advanced robotic device (Evondos E300, Evondos Telecare System, Salo, Finland). Twenty-seven home-dwelling patients retrieved their medicine sachets for 99% of the alerts. All patients and 96% of the nurses reported that the device was easy to use [7].
Line 66: rephrase the part “of they feel”
In response to your suggestion, we have made the following added as follows: Some of the desirable features that pharmacists and caregivers value include product simplicity, portability, options to lock the product, and the ability to assist with drug inventory management. These products might allow patients to independently manage their medications and could benefit highly motivated patients interested in taking control of their health and younger older adults who are more familiar with the technology [15]. However, highly motivated patients and younger older adults may not want to use such products as they have the potential to harm their dignity because these devices are locked. With the exception of Fookkun®, all medication support devices have a lock. Additionally, older people consider these devices to be difficult to refill, potentially making them cumbersome to use.
Line 72: “P-MRBs are to the physician by the patients…” A word must be missing here.
In response to your suggestion, we have made the following corrections: Patients give their P-MRBs to the physician when they visit the hospital, and the physician checks the prescription history data to ensure that there are no duplicate prescriptions. It is also given to the pharmacist when the patient visits the pharmacy, and the pharmacist checks for errors in the prescribed medications.
Line 80: Delete the abbreviation NHS. The are many abbreviations in the text and this one is not needed.
In response to your suggestion, we deleted " NHS ".
Line 97: Does this mean that patients who need medication 4 times daily cannot use the system?
In response to your suggestion, we have made the following added as follows: To dispense more than three doses a day, an additional unit allows the device to dispense up to six doses a day.
Line 106: What is a data transmitter?
In response to your suggestion, we have made the following added as follows: A prototype data transmitter was developed to retrieve Fookkun®'s medication history data.
Figure 3: Does the “x” in the middle of the figure mean that the patient hasn’t taken the medication in the middle of the day at any day?
In response to your suggestion, we have made the following added as follows: Also, if the patient is not prescribed a medication to be taken at noon, the symbol ‘x’ is displayed.
Line 133: Delete “do not have dementia” This is repetition from exclusion criteria.
In response to your suggestion, we deleted " do not have dementia ".
Line 134: Delete “patients who are not taking ODP medications, do not forget to take their medications, have full medication assistance…” This is repetition from inclusion criteria.
In response to your suggestion, we deleted “patients who are not taking ODP medications, do not forget to take their medications, have full medication assistance".
Line 153: On line 136 you write that the study was conducted from August 2022 to September 2023, here your write from August 8, 2022, to February 13, 2023. Please, describe one period at one place only.
In response to your suggestion, we deleted from the results “The experiment was conducted from August 8, 2022, to February 13, 2023.”
Line 175 Table 4. I don’t think the table is needed.
In response to your suggestion, we deleted Table 4.
Line 179 – 197: These sections are too detailed and should be shortened.
In response to your suggestion, we shortened the sentence and added Figure 4(b). “The eldest daughter called the patient's home, but the patient did not respond. A home-care nurse visited the patient's home and entered the room with a duplicate key, but the patient was not present. After receiving a report from the home-care nurse, the daughter called the neighbors and found out that the patient had fallen ill in the supermarket and was admitted to the hospital by ambulance. The eldest daughter reported that she was grateful Fookkun® allowed her to be notified of the patient's safety from afar and avoid worse outcomes.
Figure 4 shows the E-MRB data from when the patient fell ill. We subsequently checked the E-MRB by smartphone, which displayed the symbol ‘o’ on the morning of August 14th, the symbol ‘x’ from the evening of the 14th until the morning of the 16th, and the symbol ‘o’ on the evening of the 16th. No medications were taken at noon; therefore, the symbol ‘x’ is shown. We found out that the patient had left her home after the morning of the 14th and felt unwell. Additionally, we discovered that the patient was discharged and returned home during the day on August 16th. “
Line 200: It would be beneficial if the discussion started with a short resume of the findings of this study.
In response to your suggestion, we have made the following added as follows: In this study, we found that 1) the Fookkun® plus follow-up calls from a medication supporter helped prevent missed doses, improving the patient’s medication adherence, 2) the home pharmacist did not need to check for remaining medications, 3) the pharmacist and the daughter did not need to check the E-MRB, 4) the HbA1c values remained un-changed.
Line 215 – 225: Please, relates these results to your study.
In response to your suggestion, we have made the following added as follows: In another study using similar ODP medication support devices, the Evondos E300 was used for 727 days by 27 patients (mean, 26.9 d per patient). Notably, 98.7% of the alerts resulted in patients retrieving their medicine sachets on-time [7]. In another observational study, the monthly adherence rate remained >95% over the course of 6 months among the 46 participants using ‘Hello I am Spencer’ [8]. As in our study, these devices were effective in improving medication adherence.
Lie 245: Do you need to describe this that detailed?
In response to your suggestion, we deleted “Therefore, the lack of a call from Fookkun® meant that the patient was taking her medications”.
Line 298: Do not introduce a new abbreviations (DDSD) for something you only use twice.
In response to your suggestion, we have made the following corrections: "drug distribution support device".
We have corrected all that you have pointed out.
Reviewer 2 Report
Comments and Suggestions for Authors
Despite potential benefits, PHRs face challenges, such as the lack of integration between medication support devices and E-MRBs, the need for patients to manually input medication intake, and the inaccuracy of data on medication adherence. The authors present a feasibility case study of a pill dispenser (Fookkun®) and personal health records (PHRs) for verification of medication intake data. An elderly Japanese patient, her family members, and a pharmacist participated in this interview study. The study demonstrates the possible efficacy of using a pill dispenser for improving medication adherence in elderly patients. The paper is short and limited in content, but it explores an interesting question. The following improvements are suggested:
1. The rationale for the use of a pill dispenser is not fully justified. Lines 60–68 suggest that older adults may not be interested in using a pill dispenser as it is difficult to refill and also has the potential to harm their dignity. Authors may refer to the following to build their argument:
Faisal, S., Ivo, J., Abu Fadaleh, S. and Patel, T., 2023. Exploring the Value of Real-Time Medication Adherence Monitoring: A Qualitative Study. Pharmacy, 11(1), p.18.
2. Line 89-90: The following article can be cited to support the presented arguments:
Dasgupta, D., Johnson, R.A., Chaudhry, B., Reeves, K.G., Willaert, P. and Chawla, N.V., 2016. Design and evaluation of a medication adherence application with communication for seniors in independent living communities. In Amia annual symposium proceedings (Vol. 2016, p. 480). American Medical Informatics Association.
3. "Study Participants", "Evaluations" and "Study Design" can be organized under their own subheadings.
4. The authors should clarify how HbA1c values were expected to change over the course of the study. It is not clear whether it is bad for HbA1c values to stay consistent or not over the course of the study.
5. Perhaps an English version of the screen in Figure 4 could be added for improved clarity for readers.
6. The authors should explicitly discuss the interpretation of the findings in the discussion section.
Comments on the Quality of English LanguageThe paper consists of many awkward sentence structures and grammatical mistakes. Proofreading by a native English speaker can considerably improve the paper.
Author Response
Reviewer 2
Thank you for the favorable peer review. With regard to your comments, we would like to respond as follows.
- The rationale for the use of a pill dispenser is not fully justified. Lines 60–68 suggest that older adults may not be interested in using a pill dispenser as it is difficult to refill and also has the potential to harm their dignity. Authors may refer to the following to build their argument:
 We have cited the references you provided and modified them as follows:Some of the desirable features that pharmacists and caregivers value include product simplicity, portability, options to lock the product, and the ability to assist with drug inventory management. These products might allow patients to independently manage their medications and could benefit highly motivated patients interested in taking control of their health and younger older adults who are more familiar with the technology [15]. However, highly motivated patients and younger older adults may not want to use such products as they have the potential to harm their dignity because these devices are locked. With the exception of Fookkun®, all medication support devices have a lock. Additionally, older people consider these devices to be difficult to refill, potentially making them cumbersome to use.
- Line 89-90: The following article can be cited to support the presented arguments:
We have cited the references you provided and modified them as follows:Dasgupta et al. showed that the complex medical regimen maintained by many seniors frequently makes direct medication entry via an application difficult [20].
- "Study Participants", "Evaluations" and "Study Design" can be organized under their own subheadings.
In response to your suggestion, we have added a subheading: "Study Design”, "Study Participants" and "Evaluations".
- The authors should clarify how HbA1c values were expected to change over the course of the study. It is not clear whether it is bad for HbA1c values to stay consistent or not over the course of the study.
In response to your suggestion, we have made the following added in the introduction as follows: We also hypothesized that improved medication adherence would result in lower HBA1c values.
- Perhaps an English version of the screen in Figure 4 could be added for improved clarity for readers.
In response to your suggestion, we have made the following added as follows: Figure 4 (b) Screen image modified in English.
- The authors should explicitly discuss the interpretation of the findings in the discussion section.
In response to your suggestion and Reviewer 1's comments, we were corrected the discussion section.
We have corrected all that you have pointed out.
Round 2
Reviewer 2 Report
Comments and Suggestions for Authors
The authors have modified the article according to reviewers' suggestions and it is now much for readable.
Comments on the Quality of English Languagen/a